# Enzyme Immobilization on Maghemite Nanoparticles with Improved Catalytic Activity: An Electrochemical Study for Xanthine

**DOI:** 10.3390/ma13071776

**Published:** 2020-04-10

**Authors:** Massimiliano Magro, Davide Baratella, Andrea Venerando, Giulia Nalotto, Caroline R. Basso, Simone Molinari, Gabriella Salviulo, Juri Ugolotti, Valber A. Pedrosa, Fabio Vianello

**Affiliations:** 1Department of Comparative Biomedicine and Food Science, University of Padua-Agripolis, Viale dell’Università 16, 35020 Legnaro (PD), Italy; massimiliano.magro@unipd.it (M.M.); davide.baratella.1@studenti.unipd.it (D.B.); andrea.venerando@unipd.it (A.V.); nalottog@gmail.com (G.N.); 2Department of Chemistry and Biochemistry, Institute of Bioscience, Universidade Estadual Paulista, Botucatu, SP 18618-000, Brazil; caroline.basso@unesp.br (C.R.B.); valber.pedrosa@unesp.br (V.A.P.); 3Department of Geoscience, University of Padua, via G. Gradenigo 6, 35131 Padua, Italy; simone.molinari@phd.unipd.it (S.M.); gabriella.salviulo@unipd.it (G.S.); 4Regional Centre of Advanced Technologies and Materials, Palacky University in Olomouc, Slechtitelu 27, 783 71 Olomouc, Czech Republic; juri.ugolotti@upol.cz

**Keywords:** xanthine oxidases, enzyme immobilization, catalytic properties, metal nanoparticles

## Abstract

Generally, enzyme immobilization on nanoparticles leads to nano-conjugates presenting partially preserved, or even absent, biological properties. Notwithstanding, recent research demonstrated that the coupling to nanomaterials can improve the activity of immobilized enzymes. Herein, xanthine oxidase (XO) was immobilized by self-assembly on peculiar naked iron oxide nanoparticles (surface active maghemite nanoparticles, SAMNs). The catalytic activity of the nanostructured conjugate (SAMN@XO) was assessed by optical spectroscopy and compared to the parent enzyme. SAMN@XO revealed improved catalytic features with respect to the parent enzyme and was applied for the electrochemical studies of xanthine. The present example supports the nascent knowledge concerning protein conjugation to nanoparticle as a means for the modulation of biological activity.

## 1. Introduction

Hybrids constituted of nanomaterials and enzymes are promising tools for several applications, spanning from analytical devices to industrial processes. Indeed, they can represent useful building blocks for tailoring novel and specific biotechnological systems [1,2]. The preservation of the structure and biological functions of enzymes is one of the most challenging tasks for biotechnology and the fine tuning of nanoparticle–protein binding is an important issue, as it can strongly influence the catalytic properties of the final hybrid [3]. In this context, the intimate contact between protein and nanoparticle is desirable and is the aim of intense studies, for example, for developing optical and electronic devices [4].

Hybrids of nanoparticles and enzymes can count on a wide choice of core materials, ranging from metals, metal oxides to semiconductors [5]. The composition of the core determines the physical features of the final core–shell hybrid, while enzyme coatings confer the biological identity in terms of catalysis, biomimetic and recognition properties. At the same time, a variety of chemistries can be employed for binding enzymes to nanoparticle surfaces, such as physical adsorption, electrostatic interactions and covalent immobilization by glutaraldehyde or carbodiimide based reactions [6,7].

The interactions between proteins and nanomaterials are extremely complex and still far from being completely understood. In particular, the influence of the immobilization on protein structure and enzymatic activity is hardly predictable, and, within the massive body of literature dealing with enzyme immobilization, cases of enhancement of catalytic activity are rare. Nevertheless, recent studies substantiated that the improvement of enzyme activity is realistic, depending on the protein–nanoparticle combination [8,9].

Generally, nanostructured iron oxides do not represent a common option for preparing enzyme-nanomaterial hybrids. However, magnetite (Fe_3_O_4_) is the most widespread nanomaterial, even if applications involving maghemite (γ-Fe_2_O_3_) can be found [10]. Examples of enzymes successfully immobilized on iron oxide nanoparticles have been already reported such as horseradish peroxidase [11] and amino aldehyde dehydrogenase [12]. Nevertheless, the direct conjugation of enzymes to the surface of naked nanoparticles remains an open problem. Moreover, it should be considered that pristine iron oxide nanoparticles suffer for their faint colloidal stability. Hence, their surface must necessarily be engineered by proper organic or inorganic coatings for leading to aqueous colloidal suspensions and for controlling the interactions with biological macromolecules [13]. Thus, the actual nanoparticle surface is provided by the coating.

In the present work, superparamagnetic nanoparticles synthesized in house, constituting of stoichiometric maghemite (γ-Fe_2_O_3_) with a size of 10 nm, were used to create a catalytically active enzyme–nanoparticle hybrid by coupling xanthine oxidase (XO) and pristine nanoparticles. These nanoparticles, called surface active maghemite nanoparticles (SAMNs), are characterized by a specific surface behavior without any superficial modification or coating derivatization and naturally form stable colloidal suspensions in water [14]. A stable core–shell hybrid nanomaterial was obtained by mixing xanthine oxidase (the shell) with SAMNs (the core) in water, and the enzymatic activity of the immobilized XO was studied. It is noteworthy that the SAMN@XO complex displayed enhanced catalytic performances with respect to native XO. Finally, an electrochemical study for the detection of xanthine was easily fabricated by the simple magnetic immobilization of SAMN@XO on a graphite screen-printed electrode, resulting in an appealing device in comparison to already reported, more complex, xanthine detection [15]. The present research enriches the developing field on the conjugation of enzymes with nanomaterials for understanding the chemical-physical reasons behind the modulation of enzyme activity.

## 2. Materials and Methods

### 2.1. Chemicals

Chemicals were purchased at the highest commercially available purity and were used without further treatments. Iron(III) chloride hexahydrate (97%), sodium borohydride (NaBH_4_), ammonia solution (35% w/w), hydrogen peroxide (H_2_O_2_), uric acid and xanthine were obtained from Aldrich (Sigma-Aldrich, Milan, Italy). A series of Nd-Fe-B magnets (N35, 263–287 kJ/m^3^ BH, 1200mT flux density by Powermagnet—Mannheim, Germany) was used for the magnetic control of nanoparticles (magnetic driving and separations).

### 2.2. Spectrophotometric Assay for Xanthine Oxidase

The activity of xanthine oxidase (XO, from bovine milk by Sigma-Aldrich, cod. X4376) was evaluated by measuring the production rate of uric acid from xanthine at 290 nm. The reported extinction coefficient value, ε, is 9.6 × 10^3^ M^−1^ cm^−1^ [16]. In order to calculate the molar extinction coefficient (ε) of uric acid produced by the enzyme, the molar extinction coefficient (ε) of xanthine at 291 nm (3.6 × 10^3^ M^−1^ cm^−1^) was subtracted, thus obtaining a normalized molar extinction coefficient (ε) of uric acid at 291 nm of 6.0 × 10^3^ M^−1^cm^−1^. Activity measurements were carried out in 50 mM potassium phosphate buffer, pH 7.5, at 25 °C. The enzymatic activity of the native and immobilized enzymes was determined according to the Michaelis-Menten kinetic model [17].

### 2.3. Preparation of the SAMN@XO Hybrid

The synthesis of SAMNs was already described [14]. The SAMN@XO complex was prepared by adding 0.2 µM XO to a 250 µg mL^−1^ SAMN suspension, under end-over-end mixing at 4 °C for 1 h. As the enzyme binding process can be compromised by the presence of iron(III) chelating specie (e.g., phosphate ions, carboxylic acids) due to the complexation to SAMN surface, ultrapure water was used. Loosely bound enzyme was removed from the nanoparticles by repeated washing cycles, using an external magnet. The presence of the biomolecule in the supernatant was checked by the Bradford assay [18] and by assessing the corresponding enzymatic activity (see above). After three washing cycles, the activity of the immobilized enzyme was constant and no detectable enzyme release was observed in the supernatant solution (residual activity in the washing buffer was less than 1%). As for the native enzyme, the kinetic characterization of SAMN@XO (0.25 mg mL^−1^) was performed in 50 mM phosphate buffer, pH 7.5.

### 2.4. Electrochemical Measurements

SAMN@XO was magnetically immobilized on nanostructured graphite screen-printed electrodes presenting a neodymium magnet placed on the opposite side of the graphite sensing surfaces (GSPEs, Orion High Technologies S.L., Madrid, Spain). Besides the presence of graphite as working electrode, the screen-printed system included an Ag/AgCl reference electrode and a graphite counter-electrode. The electrodes were connected to a Metrohm Autolab PGSTAT128N (Metrohm Autolab B.V., Utrecht, Netherlands) driven by NOVA software (version 2.1.4, Metrohm Autolab B.V., Utrecht). Cyclic voltammetry was carried out in the −0.6 V–+0.6 V potential range at a scan rate of 0.02 V s^−1^. Electrodes were immersed in a conventional 3 mL electrochemical cell containing 50 mM phosphate buffer, pH 7.5, and 250 mM KCl as supporting electrolyte.

## 3. Results and Discussion

### 3.1. Characterization of the SAMN@XO Hybrid

The immobilization of xanthine oxidase (XO) on iron oxide nanoparticles (SAMNs) was carried out by self-assembly in water. The process was already observed with different proteins and peptides [19,20] and involves the affinity of the macromolecule for the SAMN surface.

In order to appreciate the tendency of the SAMN–XO system to spontaneously evolve into a nanohybrid, naked magnetic nanoparticles were tested for the recovery of the enzyme from the aqueous milieu: XO (0.2 µM) was incubated in the presence of increasing concentrations of SAMNs, ranging from 0.1 to 5.0 mg mL^−1^, and, after 1 h incubation, the magnetic nanomaterial was retrieved by the application of an external magnet. The enzymatic activity, under substrate saturation conditions, was used to estimate the concentration of XO in solution, and therefore to calculate the amount of bound enzyme. The amount of immobilized XO on SAMNs followed an exponential rise to a maximum value (see Figure 1A) as a function of SAMN concentration. The curve reached a plateau at around 1.0 mg mL^−1^ SAMN, corresponding to a XO/SAMNs mass ratio of 60 µg XO mg^−1^ SAMN, which was in good agreement with the protein loading reported in previous studies [20].

Transmission electron microscopy (TEM) images of SAMN@XO suggest the formation of core–shell nanostructures (see Figure 1B) displaying a massive less electron-dense phase, in which spherical objects seem to be embedded. The enzyme shell can be envisaged by the magnitude of the organic matrix, appearing, to some extent, as a bulky spacer between the iron oxide nanoparticles. Indeed, the shell thickness is compatible with the reported overall size of the XO homodimer (155Å × 90Å × 70Å, 290 kDa) [21]. It should be noted that, while SAMN@XO maintained its colloidal behavior in water, the observed aggregated character of the hybrid (Figure 1B) was induced by the drying treatment of the aqueous sample on the holey-carbon/copper-mesh grid prior the image recording by TEM. 

The SAMN@XO complex was further investigated by optical spectroscopy. The UV-Vis absorption spectrum of pristine SAMNs, acquired in water, showed a wide band with a maximum at about 400 nm characterized by an extinction coefficient of 1520 M^−1^cm^−1^ [20]. The interaction of XO with SAMN surface induced an alteration of the nanoparticle optical properties. Indeed, the spectrum of SAMN@XO was characterized by a shoulder at 400 nm with a 20% lower extinction coefficient with respect to naked SAMNs (see Figure 2). 

In order to evidence possible modifications of the enzyme catalytic behaviour upon binding, the nanostructured hybrid (SAMN@XO) was kinetically characterized.

### 3.2. Comparison of Enzymatic Activity of Native XO and SAMN@XO Hybrid

Xanthine oxidase from bovine milk is a molybdenum containing homodimeric enzyme (290 kDa) belonging to the purine degradation pathway, that catalyzes the oxidation of hypoxanthine to xanthine and consequently the conversion of xanthine into uric acid with the concomitant reduction of molecular oxygen [21,22]. It is noteworthy that the product obtained by the oxygen reduction can be hydrogen peroxide (H_2_O_2_) and superoxide anion (O_2_^−^) [23,24], and this last compound slowly dismutates into water and hydrogen peroxide [25].

Enzyme activity of native and SAMN bound XO were determined by UV-Vis spectrophotometry using xanthine as substrate as reported in the Materials and Methods section (see Figure 3).

The catalytic parameters, k_c_, K_M_ and k_c_/K_M_, of native and SAMN bound XO were determined and reported in Table 1.

Measurements were carried out by spectrophotometry at 291 nm in 50 mM phosphate buffer, using xanthine (1 µM^−1^mM) as substrate. A decrease of the enzyme affinity toward the substrate was observed upon XO immobilization to the nanoparticles. In particular, the Michaelis constant (K_M_) of the SAMN@XO hybrid was more than two times higher than that of the native enzyme. It is noteworthy that the catalytic constant (k_c_) of the SAMN@XO was 4.5 times higher than that of native XO. Therefore, catalytic efficiency (k_c_/K_M_) was two times higher in the nanoconjugate with respect to the native enzyme.

It should be mentioned that SAMN@XO maintained its catalytic activity over repeated magnetic separation and resuspension cycles, demonstrating the reusability of this magnetic drivable nano-catalyst. Interestingly, the SAMN@XO catalytic activity continued after 6 months storage at 4.0 °C. Indeed, the remaining enzymatic activity after this period was nearly 100% with respect to the freshly prepared nano-bio-conjugates.

Notably, enzyme immobilization can confer rigidity to the macromolecule, and it is generally believed that modifications of flexibility and of the local environment of enzymes affect the catalysis [26]. Actually, very often structural alterations of enzymes as a consequence of their binding to solid supports are accompanied by a decrease of their catalytic properties [27,28]. In the worst situation, the interaction with a solid surface produces a drastic reorganization of the protein leading to denaturation and to the complete loss of its biological properties [29]. Nevertheless, in the present case the integration of enzyme and nanomaterial into a hybrid improved the activity of the immobilized biomolecule, as reported in recent studies [9].

It should be recalled that the binding of proteins to SAMNs has been recently explained in terms of distribution of carboxylic groups matching the topography of under-coordinated iron(III) sites on the surface of the nanoparticle crystal domain [30]. Hence, as enzyme structure and folding are of fundamental importance for its biological activity, XO binding to SAMNs did not lead to drastic macromolecule modifications and can be likely due to the adaptation of the protein upon interaction with the SAMN surface in terms of multiple point binding.

### 3.3. Electrochemical Study of Xanthine

SAMN@XO was tested as a biological recognition element aiming at the conversion of xanthine into uric acid using a graphite screen-printed electrode (GSPE) as electrochemical transducer. The hybrid was immobilized on the electrode surface by means of a magnet positioned on the opposite side of the electrode (GPCE-SAMN@XO) as described in Methods. Specifically, GSPEs were modified by the deposition of 2 µL, 4 µL and 8 µL of a 0.25 mg mL^−1^ SAMN or SAMN@XO aqueous suspension. 

Firstly, GSPEs and SAMN modified GSPEs were electrochemically characterized by cyclic voltammetry. Control experiments were carried out as a function of xanthine and uric acid concentration. As reported in Figure 4A, neither redox peaks nor capacitive current were appreciably measured by the SAMN modified GSPE following the addition of xanthine in the electrochemical cell. On the contrary, uric acid was easily detected at both electrodes leading to an oxidation peak at a potential comprised between +0.10 V and +0.45 V vs Ag/AgCl, indicating that SAMNs do not interfere with the sensing process (see Figure 4B).

Differential pulse voltammetry was used as a sensitive electrochemical technique for the quantification of uric acid produced by the enzymatic reaction. The different applied peak potential of SWV with respect to CV measurements was likely due to the lower concentration of uric acid and to the different electrode modification (see hereafter), which resulted in negligible diffusion related issues. The modified electrodes were prepared by magnetically immobilizing the nano-hybridy dropping 2 µL, 4 µL and 8 µL of a suspension of 0.25 mg mL^−1^ SAMN@XO. The enzymatically produced uric acid was detected at +0.1 Vvs Ag/AgCl as a function of xanthine concentration, in agreement with measurements by cyclic voltammetry. The best analytical performances were obtained by depositing 2 µL of the SAMN@XO hybrid on the GPCE. The device showed a linear response in the 1–10 µM xanthine concentration range, displaying a sensitivity of 9.40 ± 0.25 × 10^−9^C µM^−1^ and a detection limit (S/N 3, according to [31]) of 0.1 µM (see Figure 5).

The effects of interferents on the electrochemical system was tested by SWV on electroactive species common in biological systems, namely reduced and oxidized nicotinamide adenine dinucleotide (NAD^+^/NADH), hydroquinone and benzoquinone, ascorbic acid and hydrogen peroxide. No detectable signal attributable to the tested electroactive substances at concentration 10 and 100 µM were observed using the described experimental conditions.

The presented SAMN@XO offering the advantage of simplicity and cost-effectiveness, displayed comparable analytical performances with respect to analogue, more complex, devices reported in literature (see Table 2). They all have advantages and disadvantages, but this new methodology seems to be the most promising. In this view, this methodology can be an interesting option for food industry issues. Indeed, biosensors for xanthine determination, as an indicator of food quality, were proposed for fish [32,33,34,35] and chicken meat [36], as well as for coffee [37].

## 4. Conclusions

In the present study, xanthine oxidase was firmly immobilized on the surface of naked maghemite nanoparticles by simple self-assembly in water, where the nanoparticle surface represents a favorable docking interface for the protein ensuring the preservation of its enzymatic activity. It is noteworthy that the as-obtained hybrid (SAMN@XO) displayed enhanced catalytic properties in comparison to the parent enzyme.

The present work provides a contribution in the study of the unpredictable properties of hybrid nanomaterials suggesting novel biomolecules-nanomaterial combinations, highlighting a case of enhancement of enzyme activity by nanoparticle conjugation.

## Figures and Tables

**Figure 1 materials-13-01776-f001:**
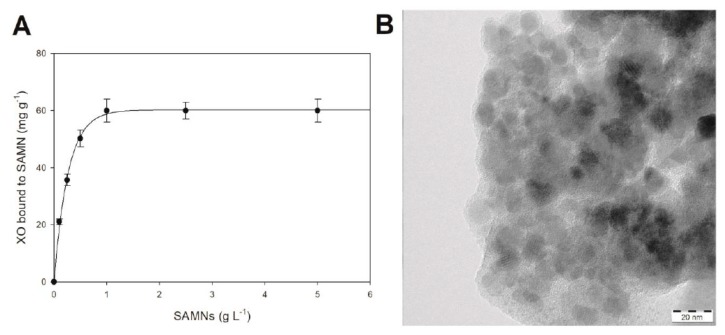
Characterization of the SAMN@XO complex. (**A**) Determination of the amount of xanthine oxidase bound on SAMNs as a function of nanoparticle concentration. (**B**) TEM image of the SAMN@XO hybrid.

**Figure 2 materials-13-01776-f002:**
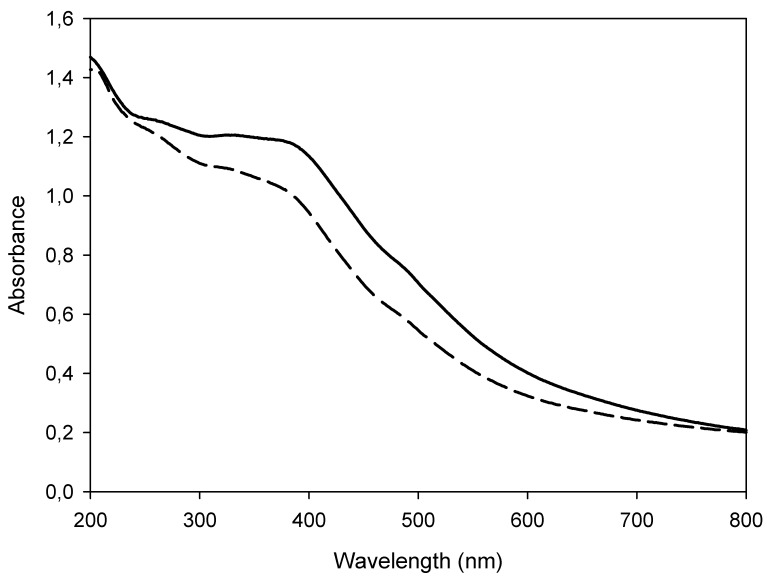
UV-Vis spectra of naked SAMNs and of the SAMN@XO complex. Measurements were carried out in water. Solid line: 0.1 g L^−1^ naked SAMNs; Dashed line: 0.1 g L^−1^ SAMN@XO.

**Figure 3 materials-13-01776-f003:**
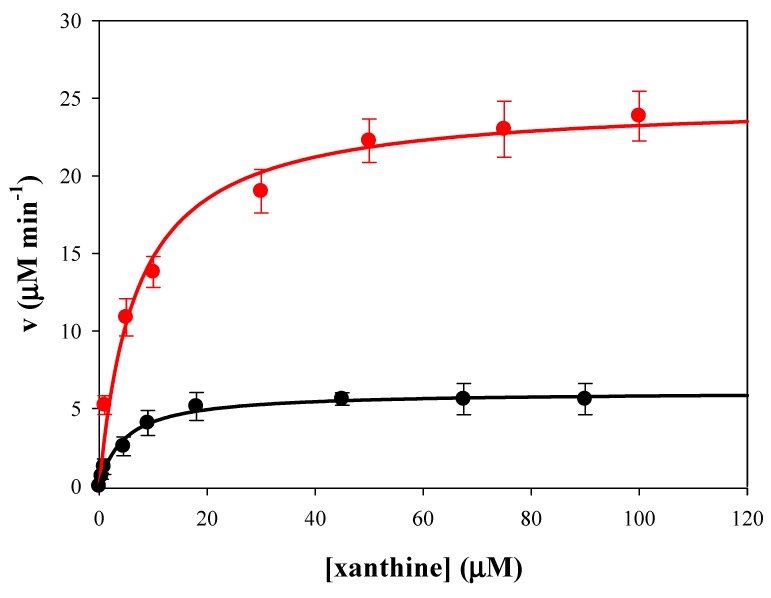
Comparison of enzymatic activity of native (black line) and immobilized Xanthine Oxidase (red line). Measurements were carried out in 50 mM potassium phosphate buffer, pH 7.5, at 25 °C in the presence of 51 nM XO (0.25 mg mL^−1^ SAMN@XO, 60 µg XO mg^−1^ SAMNs).

**Figure 4 materials-13-01776-f004:**
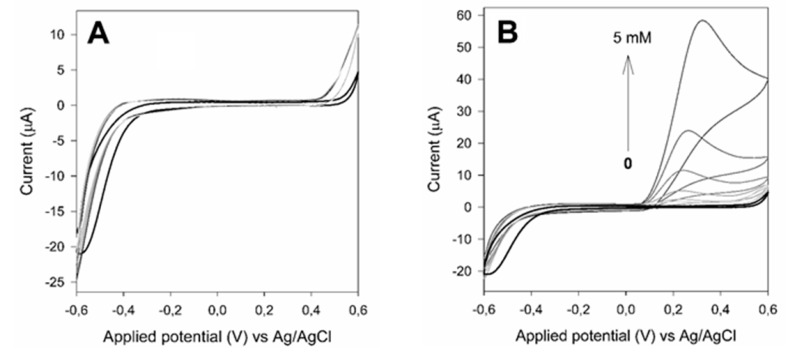
Cyclic voltammetry of graphite screen-printed electrodes (GSPEs) modified with SAMNs in the presence of xanthine (**A**) and uric acid (**B**). Measurements were carried out by dropping 2 µL of a suspension of 0.25 mg mL^−1^ SAMN in 50 mM phosphate buffer, pH 7.5. Scan speed, 20 mV s^−1^. Black line: no substrate.

**Figure 5 materials-13-01776-f005:**
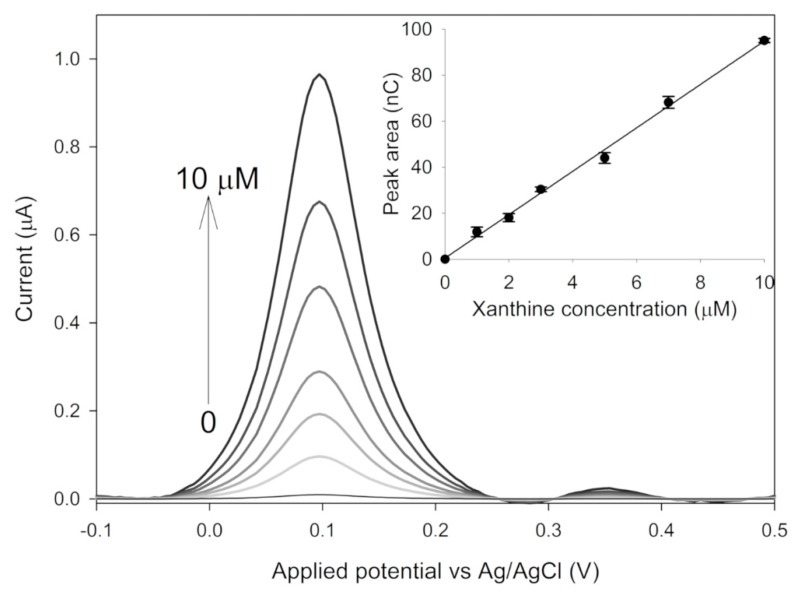
Square wave voltammetries and calibration plot obtained with SAMN@XO magnetically immobilized on a graphite screen-printed electrode as a function of xanthine concentration. Measurements were carried out in triplicate on the same sensor obtained by dropping 2 µL of a SAMN@XO suspension (0.25 mg mL^−1^) in 50 mM phosphate buffer, pH 7.5. Error bars in the inset represent standard deviations.

**Table 1 materials-13-01776-t001:** Parameters of the native XO and SAMN@XO hybrid.

Enzyme Form	Catalytic Parameters		
	K_M_ (M)	k_cat_ (min^−1^)	k_cat_/K_M_ (M^−1^min^−1^)
**Native XO**	3.1 × 10^−6^	1.1 × 10^2^	3.5 × 10^7^
**SAMNS@XO**	7.9 × 10^−6^	4.9 × 10^2^	6.2 × 10^7^

**Table 2 materials-13-01776-t002:** Comparison among different xanthine detection reported in literature.

Enzyme Support	Detection Limit (µM)	Linearity Range (µM)	Response Time (s)	Reference
Nafion	0.52	0.2–180	30	[32]
Gold nanoparticles	0.1	0.1–100	–	[33]
ZnO nanoparticles polypyrrole	0.8	0.8–40	5	[34]
Carbon nanotubes	2	2–50	150	[35]
DTP-glutaraldehyde	0.074	0.3–25	5	[36]
Fe_3_O_4_ nanoparticles Carbon nanotubes TCNQ/chitosan	0.2	1.9–230	<10	[37]
CuPtCl_6_/GC modified phospholipid membrane	20	>1	<60	[38]
SAMNs	0.1	1–10	30	This work

DTP: dithieno (3,2-b:2′,3′-d) pyrrole; TCNQ: tetracyanoquinodimethane.

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
