# Peer review of "Enzyme Immobilization on Maghemite Nanoparticles with Improved Catalytic Activity: An Electrochemical Study for Xanthine"

_materials, 2020, doi:10.3390/ma13071776_

Round 1

Reviewer 1 Report

The manuscript presents the fabrication of xanthine oxidase-caped maghemite nanoparticles. The enzymatic activity after immobilization is confirmed and proved to be comparatively higher with respect to the free enzyme. The modified particles are also used for the simple fabrication of an electrochemical xanthine biosensor. The manuscript is concise and well presented. I recommend acceptance for publication.

Author Response

Thank you for approved our work.

Reviewer 2 Report

The work has minor improvements, so it continues with the same questions for answer, so the decision must continue to improve the manuscript.

Author Response

Thank you for approved our work.

Reviewer 3 Report

The manuscript contains interesting data regarding xanthine assay and xanthine oxidase immobilization. 

The experiment should be improved for testing of selectivity (interference). In the current version, there is stated that the tested interferents were not causing a signal up to the concentration 10 uM. However, the xanthine was assayed up to the concentration 100 uM. The interferents should be tested up to the concentration 100 uM like the analyte (xanthine). The selection of interferents is not optimal. Xanthine is a purine alkaloid. Compounds structurally close to the analyte should be tested as interferents = other purine alkaloids should be chosen for the interference test. 

Author Response

The experiment should be improved for testing of selectivity (interference). In the current version, there is stated that the tested interferents were not causing a signal up to the concentration 10 uM. However, the xanthine was assayed up to the concentration 100 uM. The interferents should be tested up to the concentration 100 uM like the analyte (xanthine). The selection of interferents is not optimal. Xanthine is a purine alkaloid. Compounds structurally close to the analyte should be tested as interferents = other purine alkaloids should be chosen for the interference test. 

Following the review comments, we have made interference test using nicotinamide adenine dinucleotide (NAD+/NADH), hydroquinone and benzoquinone, ascorbic acid and hydrogen peroxide as common species. Also have tested for two concentration (10 and 100 uM). No detectable signal attributable to the tested electroactive substances at concentration 10 and 100 µM were observed using the described experimental conditions. (page 8, line 230-234).

Reviewer 4 Report

The manuscript can be accepted after major revision. My comments are,

  1. The CV result of SAMN (without enzyme) to xanthine should be included.
  2. The peak current was observed at +0.30 V in figure 4. However, the peak current is shifted to 0.10 V in DPV, why such huge deviation is observed? Comment in the manuscript.
  3. The detection limit is very good compared to other previous reports (table 1). Why such good detection limit is observed? Include comparative discussion based on rationale material properties that helped to boost detection limit.
  4. The experimental data for interference study is missing. That should be included and discussion should be included.
  5. Real sample analysis is missing.

Author Response

The CV result of SAMN (without enzyme) to xanthine should be included.

Take into consideration the referrer comments, we have include on figure 4A.

The peak current was observed at +0.30 V in figure 4. However, the peak current is shifted to 0.10 V in DPV, why such huge deviation is observed? Comment in the manuscript.

Measurements by cyclic voltammetry (CV) were carried out in an analyte concentration range orders of magnitude higher in comparison to square wave voltammetry (SWV) measurements. The two techniques were applied for different purposes: CVs were aimed at the characterization of the electrochemical system, whereas SWVs were oriented to the final application.

In this view, it is important to consider that CVs were carried out with pristine maghemite nanoparticles whereas SWV were performed with xanthine oxidase modified maghemite nanoparticles.

Faradaic current is a direct measure of the rate of the electrochemical reaction taking place at the electrode surface. The current is dependent on the rate at which material gets from the bulk of solution to the electrode and this actually concerns mass transport processes. The higher is the concentration the more problematic is the mass transport. Consequently, the peak potential will be shifted, and the effect of uric acid concentration on peak potential can be appreciated. This effect is evident in CVs presented in Figure 4 as the peak progressively shifted to higher potential as the concentration of uric acid increased. Indeed, as reported in the submitted manuscript, the redox reaction took place “between +0.10 V and +0.45 V”. In the square wave measurements reported in Figure 5,  lower concentrations were tested. Furthermore, the electrode surface was modified with maghemite nanoparticle coated by an immobilized bulky enzyme, such as xanthine oxidase. This electrode modification likely produced a layer that hinders free diffusion of molecules. In addition, due to the significant low concentration of uric acid produced by the immobilized enzyme in the presence of xanthine as substrate, the system was negligibly subjected to diffusion related issues. A brief explanation was reported in the revised text. We also included this comment in the text: “The different applied peak potential of SWV with respect to CV measurements was likely due to the lower concentration of uric acid and to the different electrode modification (see hereafter), which resulted in negligible diffusion related issues.” (page 8, line 216-219)

The detection limit is very good compared to other previous reports (table 1). Why such good detection limit is observed? Include comparative discussion based on rationale material properties that helped to boost detection limit.

Following the review comments, we have changed on the text (page 8, line 237-242).

The experimental data for interference study is missing. That should be included and discussion should be included.

Following the review comments, we have made interference test using nicotinamide adenine dinucleotide (NAD+/NADH), hydroquinone and benzoquinone, ascorbic acid and hydrogen peroxide as common species. Also have tested for two concentration (10 and 100 uM). No detectable signal attributable to the tested electroactive substances at concentration 10 and 100 µM were observed using the described experimental conditions. (page 8, line 230-234).

Round 2

Reviewer 2 Report

The manuscript was improved and could be accepted in the current form.

Reviewer 3 Report

The manuscript was corrected. I have no other comments.

Reviewer 4 Report

The revised version of the manuscript can be accepted for publication.

This manuscript is a resubmission of an earlier submission. The following is a list of the peer review reports and author responses from that submission.

Round 1

Reviewer 1 Report

The manuscript describes an interesting work on Nano-Immobilized Xanthine Oxidase with Improved Catalytic Activity: Development of a Xanthine Biosensor. The whole work must be revised on spaces and commas left. The work could be improved with several of few examples of the applicability of the biosensor.

Reviewer 2 Report

Please, check spacing words!

The line numbers for checking 'spacing words' are like these; 

21, 22, 23, 26, 27, 36, 38, 39, 44,45, 55, 56, 60, 72, 77, 83, 85, 107, 108, 121, 123, 124, 125, 170, 183, 184, 181, 183, 184, 186, 196, 197, 210, 225, 241, 242. 

Reviewer 3 Report

The manuscript presents the fabrication of xanthine oxidase-caped iron oxide nanoparticles. The enzymatic activity after immobilization is confirmed and proved to be comparatively higher with respect to the free enzyme. Moreover, the modified nanoparticles are also used for the simple fabrication of an electrochemical xanthine biosensor by magnetic entrapment over a graphite screen printed electrode.

The research is in general well presented but some points must be clarified before acceptance for publication:

- Section 3.2: in the comparison of enzymatic activity between native XO and XO at the modified nanoparticles, a clear description should be provided regarding the estimation of XO concentration used for the assay. As the authors indicate, 51 nM XO was used for this comparison. However, it is not clear how this concentration of enzyme was ensured for the assay with modified nanoparticles.

- Figure 3: the figure or caption should indicate what the two series of data presented are.

- When the authors describe the storage stability of modified nanoparticles, a more precise description should be given. It is not clear what the authors mean with "superimposable enzymatic activity". Was this measured electrochemically or by spectroscopy? Was it the same activity as for freshly prepared hybrids? It is suggested to provide a percentage of remaining activity, if possible.

- Please include an explanation on how the limit of detection was estimated.

- Figure 5: indicate the meaning of the error bars in the inset. Is it the standard deviation? How many replicates? Same or different sensors?

Reviewer 4 Report

Quite interesting manuscript but it should be improved:

  • Figure 1 should contain error bars.
  • The authors describes application of the particles as a biosensor, they provided limit of detection as a specification. However, any new analytical method should contain also validation, matrix effect testing and interference testing. 

Round 2

Reviewer 4 Report

As I commented in my previous review, the authors should provide:

  • Validation to another analytical method
  • Interference testing - is the biosensor selective to the analyte or other compounds are intereferring?

These specifications are substantial for any methods that claim applicability in analysis. 

Round 3

Reviewer 4 Report

I understand the response by the authors but I do not agree with their statement. 

I recommend to

1) either rewrite the manuscript to avoid proclamation that an analytical method is presented (= biosensor is an analytical device ) i.e. the authors are focused on technological process to prepare particles with application in a technology

2) or to make full tests that are expected to a new analytical method (validation, matrix effect, interference of compounds structurally close to the analyte etd.).

The current manuscript is not acceptable  as a work describing an analytical method.